# Anti-Obesity Effects of Tanshinone I from *Salvia miltiorrhiza* Bunge in Mice Fed a High-Fat Diet through Inhibition of Early Adipogenesis

**DOI:** 10.3390/nu12051242

**Published:** 2020-04-27

**Authors:** Dae Young Jung, Ji-Hyun Kim, Myeong Ho Jung

**Affiliations:** Division of Longevity and Biofunctional Medicine, School of Korean Medicine, Pusan National University, Yangsan 50612, Korea; dyjung999@naver.com (D.Y.J.); kimji77@pusan.ac.kr (J.-H.K.)

**Keywords:** adipogenic transcription factor, CCAAT-enhancer-binding protein β, early adipogenesis, mitotic clonal expansion, obesity, tanshinone I

## Abstract

Tanshinone I (Tan I) is a diterpenoid isolated from *Salvia miltiorrhiza* Bunge and exhibits antitumor effects in several cancers. However, the anti-obesity properties of Tan I remain unexplored. Here, we evaluated the anti-obesity effects of Tan I in high-fat-diet (HFD)-induced obese mice and investigated the underlying molecular mechanisms in 3T3-L1 cells. HFD-induced obese mice were orally administrated Tan I for eight weeks, and body weight, weight gain, hematoxylin and eosin staining and serum biological parameters were examined. The adipogenesis of 3T3-L1 preadipocytes was assessed using Oil Red O staining and measurement of intracellular triglyceride (TG) levels, and mitotic clonal expansion (MCE) and its related signal molecules were analyzed during early adipogenesis of 3T3-L1 cells. The administration of Tan I significantly reduced body weight, weight gain, and white adipocyte size, and improved obesity-induced serum levels of glucose, free fatty acid, total TG, and total cholesterol in vivo in HFD-induced obese mice. Furthermore, Tan I-administered mice demonstrated improvement of glucose metabolism and insulin sensitivity. Treatment with Tan I inhibited the adipogenesis of 3T3-L1 preadipocytes in vitro, with this inhibition mainly occurring at an early phase of adipogenesis through the attenuation of MCE via cell cycle arrest at the G1/S phase transition. Tan I inhibited the phosphorylation of p38, extracellular signal-regulated kinase (ERK), and Akt during the process of MCE, while it stimulated the phosphorylation of AMP-activated protein kinase. Furthermore, Tan I repressed the expression of CCAAT-enhancer-binding protein β (*C/EBPβ*), histone H3K9 demethylase *JMJD2B*, and subsequently cell cycle genes. Moreover, Tan I regulated the expression of early adipogenic transcription factors including GATAs and Kruppel-like factor family factors. These results indicate that Tan I prevents HFD-induced obesity via the inhibition of early adipogenesis, and thus improves glucose metabolism and insulin sensitivity. This suggests that Tan I possesses therapeutic potential for the treatment of obesity and obesity-related diseases.

## 1. Introduction

Obesity causes a critical health problem because of an increase in the incidence of metabolic diseases including type 2 diabetes mellitus, atherosclerosis, and hypertension. Energy imbalance between energy uptake and energy consumption leads to the storage of excessive energy in the adipose tissue, resulting in obesity. The expansion of the adipose tissue results from hyperplasia and hypertrophy [1]. Hyperplasia is induced by the increase of adipocyte numbers due to differentiation (adipogenesis), and hypertrophy is induced by the increase of adipocyte size due to triglyceride (TG) accumulation [1]. Therefore, the prevention of adipogenesis is a candidate therapeutic approach for the development of anti-obesity agents.

Adipogenesis is mediated by various regulators including transcriptional factors, signaling kinases, and lipid metabolic enzymes [2]. The molecular and cellular processes of adipogenesis have been characterized, and widely studied, in 3T3-L1 cells. The adipogenesis of 3T3-L1 preadipocyte is composed of two stages, an early stage from 0 to 48 h after adding differentiation medium, and a late stage from day 2 to day 8. In the early stage, growth-arrested confluent 3T3-L1 preadipocytes re-enter the cell cycle simultaneously and go through two rounds of the cell cycle (which is called mitotic clonal expansion (MCE)) [3]. MCE is a critical step for terminal differentiation into mature adipocytes [3]. The mitogen-activated protein kinase (MAPK) pathways, including extracellular signal-regulated kinase (ERK) and p38 MAPK, play a positive role in the MCE process during early adipogenesis [4]. Cyclin and cyclin-dependent kinase (CDK) regulate the progression of MCE, through the regulation of cell division cycles [3]. The activated MCE proceeds to the late phase of adipogenesis, during which well-programmed transcription cascades stimulate the expression of terminal differentiation-related genes including fatty acid synthase (*FAS*), fatty acid binding protein 2 *(aP2*), and fatty acid translocase/cluster of differentiation 36 (*FAT/CD36*) [5].

The transcriptional cascades of adipogenesis include peroxisome proliferator-activated receptor γ (PPARγ), CCAAT-enhancer-binding protein α, C/EBPβ, C/EBPδ, and sterol regulatory element-binding proteins 1c (SERBP1c). C/EBPβ and C/EBPδ, upregulated at the early stage, stimulate the expression of cell cycle-associated genes and terminal transcriptional factors such as *PPARγ2* and *C/EBPα* during MCE process [5]. Then, PPARγ and C/EBPα stimulate the expression of lipid metabolism genes for the formation of mature adipocytes [5].

*Salvia miltiorrhiza* Bunge (Danshen) is a medicinal herb, traditionally used for the treatment of several diseases including cancer, hyperlipidemia, and cerebrovascular disease [6]. Tanshinone I (Tan I) is one of the major diterpene compounds (a diterpenoid) isolated from *S. miltiorrhiza* Bunge and exhibits a wide spectrum of antitumor effects in several cancers including gastric, prostate, and breast cancer [6]. However, the anti-obesity effects of Tan I have remained unexplored. Thus, in the current study, the protective effects of Tan I against diet-induced obesity (DIO) were investigated in high-fat-diet (HFD)-induced obese mice, and the underlying mechanisms, involved in the anti-obesity effects exerted by Tan I, were characterized in 3T3-L1 cells. Here, for the first time, we revealed that Tan I exerted an anti-obese effect against DIO mice, through the inhibition of early adipogenesis, via suppression of MCE and regulation of the early adipogenic transcription cascade.

## 2. Materials and Methods

### 2.1. Reagents

Tanshinone I (≥98% purity) was purchased from ChemFaces (Wuhan, China). Dulbecco’s modified Eagle’s medium (DMEM), bovine calf serum (BCS), fetal bovine serum (FBS), and penicillin/streptomycin were obtained from HyClone (Logan, UT, USA). 3-isobutyl-1-methylxanthine, rosiglitazone, dexamethasone, and insulin were purchased from Sigma-Aldrich (St. Louis, MO, USA). Anti-p38, anti-p42/44 ERK, anti-pAkt, and anti-pAMPK were obtained from Cell Signaling Technology (Danvers, MA, USA). Antibodies against PPARγ, FAS, aP2, Cyclin D2, Cdk2, p21, p27, and β-actin were obtained from Santa Cruz Biotechnology (Santa Cruz, CA, USA). The reagent for the measurement of TG levels was from Asan Pharmaceutical Co., Ltd. (Seoul, South Korea).

### 2.2. Animal Study

Male 6-week-old C57BL/6 mice were obtained from Jung-Ang Lab Animal, Inc. (Seoul, South Korea). All mice were fed a normal diet (ND) or an HFD for four weeks. Then, the HFD-fed mice were randomly divided into four groups (*n* = 10/group): HFD (vehicle-treated) group, HFD plus low-dose Tan I (2 mg/kg of body weight) group, HFD plus high-dose Tan I (5 mg/kg of body weight) group, and metformin (200 mg/kg body weight) group as a positive control. AIN93G diet with the high-fat diet containing 60% kcal fat was used, and the control diet contained 10% kcal fat. Tan I was dissolved in DMSO at 1 mg/mL, diluted in distilled water, and administered orally three times a week, for eight weeks. The animal experiments were approved by the Animal Care and Use Committee at Pusan National University in accordance with the established ethical and scientific care procedures (approval number: PNU-2019–2203).

### 2.3. Glucose and Insulin Tolerance Tests

To perform the intraperitoneal glucose tolerance test (IPGTT), mice were fasted for 12 h with access to drink water. All mice were weighed and then injected intraperitoneally with glucose (2 g/kg body weight). Blood glucose levels were measured by placing on the strip of glucose meter at 30, 60, 90, and 120 min thereafter. The insulin tolerance test (ITT) was conducted after mice were fasted for 6 h, and all mice were injected with human insulin (0.75 U/kg body weight) intraperitoneally. Glucose levels were measured using a glucometer at 15, 30, 45, 60, 90, and 120 min.

### 2.4. Analysis of Serum Biological Parameters

Whole blood was collected in a heparinized tube and centrifuged at 3000 rpm for 20 min at 4 °C. The concentrations of plasma free fatty acid (FA) and total cholesterol level were measured by colorimetric methods using kits (Biomax, Seoul, South Korea), and plasma TG levels were measured by commercial spectrophotometric kits (Asan Pharmaceutical Company, Seoul, South Korea). All plasma samples were aliquoted at −80 °C before the measurements.

### 2.5. Cell Culture and Adipocyte Differentiation

The 3T3-L1 mouse preadipocytes obtained from American Type Culture Collection (Manassas, VA, USA) were maintained in DMEM supplemented with 10% (v/v) BCS and penicillin/streptomycin (100 U/mL/100 mg/mL). Post-confluent 3T3-L1 cells were differentiated in differentiation medium (MDI) containing DMEM, 0.5 mM IBMX, 10% FBS, 5 μg/mL insulin, 2 mM rosiglitazone, and 1 μM dexamethasone as described previously [7].

### 2.6. Cell Proliferation Assays

The 3T3-L1 cell proliferation was measured using the Cell Proliferation Kit II (XTT) (Roche Diagnosis GmbH, Penzberg, Germany) according to the manufacturer’s instructions.

### 2.7. Oil Red O (ORO) Staining

The 3T3-L1 cells were washed with phosphate-buffered saline (PBS) and fixed with 4% formaldehyde for 30 min. After discarding the formalin solution, the cells were washed with PBS and stained with Oil Red O, as described previously [7].

### 2.8. Measurement of TG Levels

The 3T3-L1 preadipocytes were washed with PBS, and the harvested cell suspensions were extracted with chloroform:methanol (3:1, v:v) for 1 h. The TG levels in the mixture were measured as described previously [7].

### 2.9. Quantitative Polymerase Chain Reaction (qPCR)

Total RNA was extracted from 3T3-L1 cells using TRIzol reagent (Invitrogen Life Technologies, Carlsbad, CA, USA) according to the manufacturer’s instructions. cDNA synthesis was made by TOPscript^TM^ RT DryMIX (Enzynomics, Daejeon, South Korea), and subjected to qPCR using a SYBR Green master mixture with gene-specific primers (Appendix A).

### 2.10. Western Blotting

The proteins were prepared from 3T3-L1 cells using Pro-prep^TM^ protein extraction solution (Intron Biotechnology, Seongnam, South Korea) and assayed, as described previously [7].

### 2.11. Cell Cycle Analysis

The 3T3-L1 cells were collected and fixed with 70% ice-cold ethanol for 12 h at 4 ℃. After washing with ice-cold PBS, the cells were stained with 25 μg/mL propidium iodide (PI) (BioLegend, San Diego, CA, USA) and analyzed by FACS (fluorescence activated cell sorter) Canto II (Becton, Dickins and Company, San Jose, CA, USA) according to the manufacturer’s instructions. Analysis of the cell cycle populations was conducted using the BD Pro software (BD Biosciences, San Jose, CA, USA).

### 2.12. Statistical Analysis

All data were expressed as the mean ± SEM. The statistical analysis was conducted using one-way ANOVA analysis of variance followed by Tukey’s test. Differences were considered statistically significant at *p* values <0.05.

## 3. Results

### 3.1. Tan I Prevented HFD-Induced Obesity and Promoted Glucose Utilization and Insulin Sensitivity

To assess the anti-obesity effects of Tan I, we performed in vivo experiments using HFD-fed C57BL/6J mice. Six-week-old C57BL/6J mice were fed HFD or ND for 4 weeks, and the HFD mice were further administered with a low dose (2 mg/kg of body weight) or high dose (5 mg/kg of body weight) of Tan I, or metformin (200 mg/kg body weight) as a positive control, for 8 weeks. No significant differences in food intake were observed between the groups (data not shown). Mice fed HFD for 12 weeks showed increased body weight, compared to the ND mice (Figure 1A). However, the administration of both low dose and high dose Tan I significantly reduced body weight (Figure 1A) and prevented weight gain (Figure 1B) consistent with metformin-fed mice. At the end of the experiment, the average weight of HFD-induced obese mice was 46.1 ± 1.5 g, while the average weight of HFD mice administered low dose Tan I, high dose Tan I, and metformin was 40.1 ± 1.3 g, 40.5 ± 1.1 g, and 37.9 ± 1.1 g, respectively. Additionally, photography revealed that HFD mice demonstrated an increased size of the whole body and white adipose tissue, compared to ND mice (Figure 1C). However, administration with both doses of Tan I decreased the whole body and white adipose tissue size (Figure 1C). Hematoxylin and eosin (H&E) staining also showed that HFD feeding increased the average adipocyte size in the white adipose tissue, compared to ND mice (Figure 1D); however, Tan I administration to HFD obese mice reduced adipocyte enlargement (Figure 1D). These results indicated that Tan I could potentially prevent the HFD-induced body weight gain and adiposity. Next, we examined the blood biochemical profiles of HFD-induced obese mice to confirm the anti-obesity effects of Tan I. HFD feeding resulted in higher blood levels of glucose (Figure 1E), free fatty acid (Figure 1F), total TG (Figure 1G), and total cholesterol (Figure 1H). However, mice fed with low and high doses of Tan I demonstrated reduced blood levels of these parameters (Figure 1E–H), similar to results observed in metformin-treated mice. These results indicated that Tan I could possibly improve glucose and lipid homeostasis in HFD-induced obese mice. To confirm the improved effect of Tan I on glucose metabolism and insulin sensitivity by the prevention of obesity, we performed glucose and insulin tolerance tests in mice. As shown in Figure 2A, Tan I-treated HFD-induced obese mice demonstrated a higher reduction in glucose following glucose loading, at specified time points, compared to HFD-induced obese mice not treated with Tan I. Furthermore, we investigated the beneficial effects of Tan I on HFD-induced insulin resistance using the insulin tolerance test. As shown in Figure 2B, Tan I-treated HFD-induced obese mice demonstrated a higher sensitivity to insulin administration, compared to HFD-induced obese mice not treated with Tan I. Taken together, these results indicated that Tan I significantly prevented HFD-induced obesity and improved glucose utilization and insulin sensitivity.

### 3.2. Tan I Inhibited Early Adipogenesis of 3T3-L1 Preadipocytes

The prevention of adipocyte differentiation (adipogenesis) is a candidate therapeutic target against obesity [1]. In order to elucidate the molecular mechanisms involved in the anti-obesity effects of Tan I, we first examined the inhibitory effects of Tan I on the adipogenesis of 3T3-L1 preadipocytes. The 3T3-L1 preadipocytes were differentiated in MDI medium with the indicated concentrations of Tan I, or without Tan I, for 8 days. The ORO staining revealed that treatment with 5 or 10 μM Tan I inhibited the adipogenesis of 3T3-L1 preadipocytes in a dose-dependent manner (Figure 3A). Furthermore, the measurement of the intracellular TG levels revealed that treatment with Tan I resulted in a significant reduction of intracellular TG levels (Figure 3B), consistent with results of ORO staining. To confirm Tan I-mediated inhibition of adipogenesis, we measured protein levels of adipogenic markers, PPARγ and its target genes (*aP2* and *FAS*). Western blotting revealed that Tan I treatment lowered protein levels of PPARγ, aP2, and FAS (Figure 3C). Taken together, these results indicated that Tan I inhibited the adipogenesis of 3T3-L1 preadipocytes.

Adipogenesis is divided into two phases, an early phase from day 0 to day 2, and a late phase from day 3 to day 8 [2]. To determine the adipogenic phase sensitive to the anti-adipogenic effect of Tan I, 3T3-L1 preadipocytes were incubated with Tan I, at various time points after treatment with MDI, as illustrated in Figure 3D, and adipogenesis levels were examined by ORO staining. As shown in Figure 3E, the treatment with Tan I at the early adipogenic phase (days 0–2 and 0–4) inhibited adipogenesis significantly, and this effect was very similar to that obtained with continuous treatment (days 0–8) (Figure 3E). However, the treatment with 10 μM Tan I after day 2 (days 2–4 and 4–8) demonstrated a lowered inhibitory effect (Figure 3E). These results were also confirmed by the measurement of intracellular TG content (Figure 3F). Taken together, these results demonstrated Tan I-mediated inhibition of adipogenesis is primarily due to its action at the early adipogenesis phase.

### 3.3. Tan I Inhibited MCE during the Early Adipogenesis Phase

The incubation of growth-arrested 3T3-L1 preadipocytes with MDI causes MCE during early adipogenesis [3]. To determine whether Tan I affected MCE, we examined the effects of Tan I on the proliferation of 3T3-L1 cells during the MCE process. As shown in Figure 4A, incubation of 3T3-L1 preadipocytes with MDI resulted in increased cell numbers; however, treatment with 10 μM Tan I significantly reduced the cell numbers during 48 h, suggesting that Tan I inhibited the MCE process during the early adipogenesis phase. We further assessed whether Tan I-inhibition of MCE is mediated by the changes in cell cycle progression during the MCE process. To this end, we examined the cell cycle distribution in Tan I-treated 3T3-L1 cells using flow cytometric analysis. Incubation of 3T3-L1 preadipocytes with MDI decreased the cell population in the G0/G1 phase, and increased the cell population in the S phase (Figure 4B), indicating that the induction of adipogenesis with MDI leads to normal cell cycle progression from the G0/G1 phase to the S and G2/M phases. However, 10 μM Tan I treatment markedly increased the G0/G1 cell population, and subsequently decreased the cell numbers in the S and G2/M phases (Figure 4B). These results demonstrated that Tan I could arrest the cell cycle at the G0/G1 phase during the MCE process and inhibit cell cycle progression into the S phase.

Cell cycle progression is regulated by the activation of CDKs and their associated cyclins [3]. To determine the effects of Tan I on the expression of CDKs and cyclins, we examined their protein and mRNA levels in Tan I-treated 3T3-L1 cells. As shown in Figure 4C, western blotting revealed that MDI incubation increased the protein levels of cyclin A, cyclin D2, and Cdk2, at 24 h; however, treatment with Tan I blocked these levels in a dose-dependent manner. Additionally, qPCR assay also revealed that Tan I treatment blocked the MDI-induced mRNA levels of *cyclin A*, *cyclin D2*, and *Cdk2* at 18 h (Figure 4D). Furthermore, we examined the expression of cell cycle inhibitors in Tan I-treated 3T3-L1 cells. Western blotting revealed that Tan I enhanced the protein levels of p21 and p27, inhibitory regulators of the cell cycle, 24 h after MDI treatment (Figure 4C). These results indicated that Tan I inhibited the MCE process through the downregulation of cyclin A, cyclin D, and Cdk2, and upregulation of p21 and p27, which may contribute to a delayed entry into the S and G2 phases of the cell cycle, and subsequent growth inhibition.

### 3.4. Tan I Inhibited MAPK and PI3K/Akt Signaling and Activated AMPK

The MAPKs and the PI3K/Akt signaling pathways play an important role in the control of cell proliferation, survival, and differentiation [4]. To further elucidate the mechanism underlying Tan I-mediated inhibition of early adipogenesis, we evaluated the effects of Tan I on cell cycle signaling pathways including p38, ERK, and Akt in Tan I-treated 3T3-L1 adipocytes. As shown in Figure 5A, treatment with Tan I reduced the phosphorylation of p38, p-42/44 (ERK), and Akt in dose- and time-dependent manners. These results suggested that the suppression of p38, p-42/44 (ERK), and PI3K/Akt signaling may be involved in the Tan I-mediated inhibition of the MCE process. Furthermore, AMP-activated protein kinase (AMPK) activation inhibits the adipogenesis of 3T3-L1 preadipocytes through suppression of MCE and downregulation of several adipocyte-specific transcription factors [8]. Thus, we investigated whether Tan I activated AMPK in 3T3-L1 adipocytes. As shown in Figure 5B, Tan I treatment increased the phosphorylation of AMPK in 3T3-L1 adipocytes in dose- and time-dependent manners. Taken together, these results suggested that AMPK activation may also be involved in the Tan I-mediated inhibition of the MCE process.

### 3.5. Tan I Repressed the Expression of Early Adipogenic Transcription Factor C/EBPβ, Its Target Histone Demethylase JMJD2B, and Cell Cycle Genes during the MCE Process

C/EBPβ, an early adipogenesis transcription factor which is expressed within 4 h after treatment with MDI, is essential for the MCE process through regulation of cell cycle genes [9]. To further investigate whether Tan I regulates the expression of C/EBPβ during early adipogenesis, we examined the expression of C/EBPβ in Tan I-treated 3T3-L1 cells. As shown in Figure 6A, western blotting revealed that MDI incubation increased the C/EBPβ level; however, Tan I treatment decreased the protein level in a dose-dependent manner. Additionally, qPCR revealed that *C/EBPβ* mRNA level was increased by the MDI incubation; however, Tan I treatment significantly decreased the mRNA level during the MCE process (Figure 6A). Reportedly, MDI-induced C/EBPβ subsequently stimulates the expression of PPARγ and C/EBPα, which are necessary for terminal adipogenesis [9]. As shown in Figure 6A, western blotting demonstrated that Tan I-mediated repression of C/EBPβ caused a decrease in the protein levels of PPARγ and C/EBPα (Figure 6A).

Recently, it was reported that C/EBPβ stimulates the expression of histone H3K9 demethylase JMJD2B and C/EBPβ target cell cycle genes, including *Cdc25c*, *Cdc45l*, and *Mcm3* [10]. We further investigated whether Tan I regulated the epigenetic enzyme JMJD2B and cell cycle genes concomitant with decreased C/EBPβ during the MCE process. As shown in Figure 6B, Tan I treatment reduced mRNA levels of *JMJD2B*, along with the decreased *C/EBPβ*, and subsequently repressed the expression of cell cycle genes including *Cdc25c*, *Cdc45l*, and *Mcm3*, at the indicated time points during the MCE process (Figure 6C). Taken together, these results suggested that Tan I-inhibition of MCE might be epigenetically regulated through downregulation of C/EBPβ, JMJD2B, and cell cycle genes.

### 3.6. Tan I Downregulated Early Adipogenic Activators, While Upregulating Early Adipogenic Inhibitors

To further characterize the mechanism involved in Tan I-mediated inhibition of adipogenesis, we also detected the effects of Tan I on the expression of several early adipogenic transcription factors in 3T3-L1 cells treated with MDI. qPCR assay revealed that Tan I repressed the expression of early adipogenic activators, *SREBP1c*, *KLF* (Krüppel-like factor) *4*, and *KLF6* (Figure 7A), whereas it increased the expression of early adipogenic repressors, *KLF2*, *GATA* (GATA binding protein) *2*, and *GATA3* (Figure 7B).

## 4. Discussion

It has been reported that some phytochemicals inhibited adipogenesis and thereby exhibited anti-obesity effects [11]. Tan I, a diterpenoid isolated from *S. miltiorrhiza* Bunge, is reported to have antitumor effects in several cancers [6]. However, the anti-obesity effects of Tan I have not been investigated. Thus, in the current study, we evaluated the anti-obesity properties of Tan I in HFD-induced obese mice and characterized its underlying mechanisms in 3T3-L1 cells. First, we investigated whether Tan I exerted in vivo anti-obesity effects in HFD-induced obese mice. Administration of Tan I to HFD-induced obese mice lowered HFD-induced body weight, weight gain, and white adipocyte size, with no significant change in the food intake. Further, Tan I administration also reduced HFD-induced blood levels of glucose, free fatty acids, total TG, and total cholesterol, indicating that Tan I can prevent the HFD-induced obesity and ameliorate the serum metabolic parameters. Moreover, Tan I improved HFD-induced glucose and insulin tolerance. Based on these results, Tan I has a great potential in the prevention and treatment of obesity and the improvement of glucose and lipid metabolism.

Next, we investigated the molecular mechanisms involved in the protection afforded by Tan I against HFD-induced obesity. Adipocyte hyperplasia contributes to obesity by increasing the numbers of adipocytes, via differentiation of preadipocytes into mature adipocytes (adipogenesis) [1]. Therefore, inhibition of adipogenesis is a potential target of anti-obesity agents. On the basis of this knowledge, we evaluated whether Tan I inhibited adipogenesis of 3T3-L1 preadipocytes. Our current results revealed that treatment with Tan I significantly inhibited MDI-induced adipogenesis of 3T3-L1 preadipocytes, Furthermore, the Tan I-mediated inhibition of adipogenesis mostly occurred during the early adipogenesis phase, since Tan I treatment at the early adipogenic phase significantly inhibited adipogenesis, almost similar to the effect of a continuous treatment.

During the early adipogenesis phase, incubation of the growth-arrested preadipocytes with MDI causes the preadipocytes to process MCE [3]. Then, MCE activates early adipogenic transcription factors including C/EBPβ/δ during early adipogenesis, which subsequently stimulate the late adipogenic transcription factors, including PPARγ and C/EBPα, responsible for the induction of terminal adipocyte differentiation [3]. Therefore, to further investigate whether Tan I inhibited the MCE process during early adipogenesis, we assessed the effects of Tan I on the proliferation and cell cycles of 3T3-L1 cells during this phase. The current data revealed that Tan I treatment inhibited cell proliferation of 3T3-L1 adipocytes and arrested cell-cycle from G1/S into the S and G2/M phases during MCE. This suggested that Tan I inhibited the cell cycle pathway, resulting in the suppression of MCE at the early adipogenesis phase. In addition, our current results revealed that Tan I treatment inhibited the expression of cyclin A, cyclin D2, and Cdk2 during the MDI process and stimulated the expression of p21 and p27, which are negative regulators of the G1/S transition of the cell cycle. The cyclin A- and E-dependent activation of Cdk2, and downregulation of p27 are required for the progression of the cell cycle from G1 to S phase [3]. During MCE, cyclin D and cyclin E are activated, and the activated cyclin D interacts with Cdk4 and Cdk6, consequently inducing the G1/S transition. Meanwhile, cyclin E interacts with Cdk2 which recruits cyclin A, and the association of Cdk2 and cyclin A are essential for G1/S progression. Therefore, cyclins A and D are essential for the progression of the cell cycle from the G0/G1 phase to the S and G2/M phases. Based on our current results, the downregulation of cyclin A, cyclin D2, Cdk2, and upregulation of p21 and p27 may lead to the Tan I-mediated inhibition of MCE, through the delayed entry of G0/G1 cells into the S phase.

The MAPKs including p38, ERK, and PI3/Akt pathways play important roles in differentiation, proliferation, mitosis, and cell survival [12]. The MAPKs are activated during the MCE process and stimulate MCE through the regulation of cyclin D and p27 expression [4]. Furthermore, the MAPKs and PI3/Akt pathways activate C/EBPβ and its target mitogenic signaling molecules [13]. Therefore, we investigated whether Tan I could affect MAPKs and PI3K/Akt signaling pathways during the MCE process. Our current results revealed that treatment with Tan I reduced the phosphorylation of p38, ERK, and Akt in dose- and time-dependent manners in 3T3-L1 adipocytes treated with MDI, indicating that Tan I may inhibit the cell proliferation during MCE through the suppression of MAPKs and PI3/Akt pathways. Notably, AMPK inhibits MCE through suppression of mammalian target of rapamycin complex 1 (mTORC1) signaling and stimulates Wnt/β-catenin signaling, which results in the repression of PPARγ expression [8,14]. Furthermore, AMPK attenuates TG accumulation in adipocytes during adipogenesis via deactivation of acetyl coA carboxylase, fatty acid synthesis enzymes [15]. Therefore, we further investigated whether Tan I could activate AMPK in 3T3-L1 cells. Treatment with Tan I enhanced the phosphorylation of AMPK in dose- and time-dependent manners in 3T3-L1 cells treated with MDI, which suggested that Tan I-activated AMPK might also contribute to inhibiting MCE.

C/EBPβ plays a critical role in the MCE process through the stimulation of cell cycle genes [9]. Recently, it was reported that C/EBPβ increases the expression of histone H3K9 demethylase, JMJD2B, which upregulates C/EBPβ target cell cycle genes, including *Cdc25c*, *Cdc45l*, and *Mcm3*, as a cofactor of C/EBPβ via removing inhibitory histone marker H3K9me3 in their promoter regions [10]. Thus, we considered that the Tan I inhibition of MCE might be regulated epigenetically through the downregulation of C/EBPβ–JMJD2B, and cell cycle genes. Our current data demonstrated that treatment with Tan I reduced the expression of C/EBPβ and its target epigenetic histone demethylase JMJD2B, and thereby repressed the expression of C/EBPβ-regulated cell cycle genes *Cdc25c*, *Cdc45l*, and *Mcm3* during MCE process. These results indicated that Tan I inhibited MCE epigenetically through the downregulation of C/EBPβ, JMJD2B, and cell cycle genes.

Adipogenesis is tightly controlled by transcription factor cascades, which activate or repress adipogenesis transcription factors of each other [5]. In the current study, we found that Tan I downregulated early adipogenic activators, SREBP1c, KLF4, and KLF6, whereas it upregulated early adipogenic repressors, KLF2, GATA2, and GATA3, suggesting that Tan I may also inhibit adipogenesis through the regulation of adipogenesis transcription factor cascades during the early adipogenesis process. Besides Tan I, two compounds belonging to the group of tanshinones including tanshinone IIA and cryptotanshinone have been reported to exert anti-adipogenic effects through the downregulation of positive adipogenesis factors including C/EBPα, C/EBPβ, and PPARγ; upregulation of negative adipogenesis factors GATA2, CHOP (C/EBP homologous protein), and TNF-α (tumor necrosis factor); and inhibition of STAT3/ 5 signaling [16,17]. The important difference in molecular mechanisms between Tan I and the other two compounds is that Tan I attenuates the MCE process during early adipogenesis and represses the expression of epigenetic regulator, JMJD2B associated with activation of adipogenesis, which may also contribute to the inhibition of adipogenesis of 3T3-L1.

## 5. Conclusions

Tan I prevented HFD-induced obesity via the inhibition of early adipogenesis and thus ameliorated glucose metabolism and insulin sensitivity. Tan I-mediated inhibition of early adipogenesis was through the impairment of MCE via arresting the cell cycle. Inhibition of ERK, p38, and PI3K/Akt signaling and activation of AMPK pathway could be involved in the inhibition of MCE. In addition, downregulation of C/EBPβ, JMJD2B, and cell cycle genes and regulation of early adipogenic transcription factors also contribute to inhibition of early adipogenesis.

## Figures and Tables

**Figure 1 nutrients-12-01242-f001:**
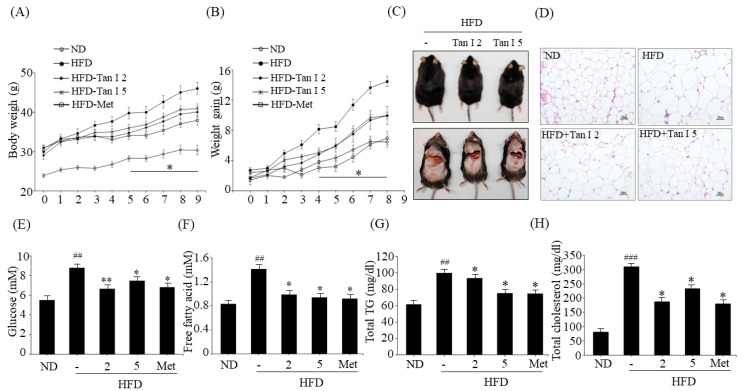
Tanshinone I (Tan I) prevented high-fat-diet (HFD)-induced obesity and ameliorated blood biochemical parameters in mice. (**A**) Body weight. (**B**) Weight gain. (**C**) Photograph of whole body and visceral adipose tissues. (**D**) Hematoxylin and eosin (H&E) staining (scale bar, 500 μm). Blood levels of glucose (**E**), free fatty acid (**F**), total TG (**G**), and total cholesterol (**H**) in HFD obese mice. The data are presented as the mean ± SEM of ten mice. ^##^
*p* < 0.01, ^###^
*p* < 0.001 vs. normal diet (ND)-fed mice, * *p* < 0.05, ** *p* < 0.01 vs. HFD-fed mice. Met means metformin administration.

**Figure 2 nutrients-12-01242-f002:**
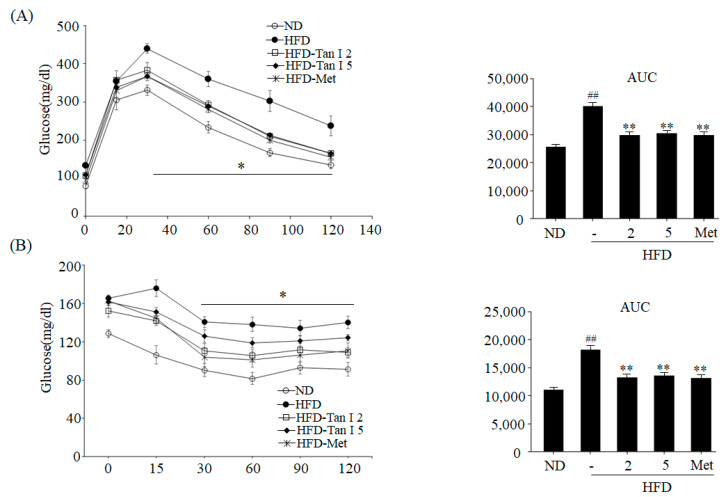
Tan I improved glucose and insulin tolerance in HFD obese mice. (**A**) Intraperitoneal glucose tolerance test (IPGTT) and the glucose area under the curve (AUC) during IPGTT. (**B**) Insulin tolerance test (ITT) and the glucose inverse AUC during ITT. The data are presented as the mean ± SEM of ten mice. ^##^
*p* < 0.01 vs. ND-fed mice, * *p* < 0.05, ** *p* < 0.01 vs. HFD-fed mice. Met means metformin administration.

**Figure 3 nutrients-12-01242-f003:**
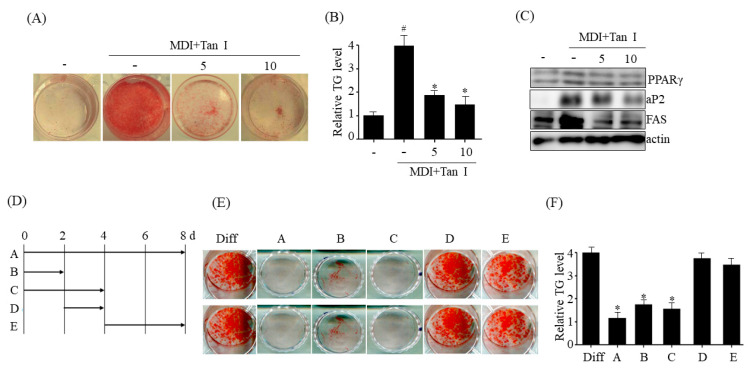
Tan I inhibited early adipogenesis of 3T3-L1 preadipocytes. The 3T3-L1 preadipocytes were differentiated in the differentiation medium (MDI) with 5 or 10 μM Tan I for 8 days, and the adipogenesis was determined by Oil Red O (ORO) staining (**A**), intracellular triglyceride (TG) levels (**B**) and the protein levels of adipogenic markers including *PPARγ*, *aP2*, and *FAS* (**C**). The full length western blots are provided in Appendix A. The 3T3-L1 preadipocytes were differentiated in the MDI with 10 μM Tan I on indicated periods (**D**). The adipogenesis was assessed by ORO staining (**E**) and intracellular TG levels (**F**) on indicated periods. The data are presented as the mean ± SEM of three replicate experiments. ^#^
*p* < 0.05 vs. undifferentiated 3T3-L1 preadipocytes, * *p* < 0.05 vs. differentiated 3T3-L1 adipocytes without Tan I treatment (Diff).

**Figure 4 nutrients-12-01242-f004:**
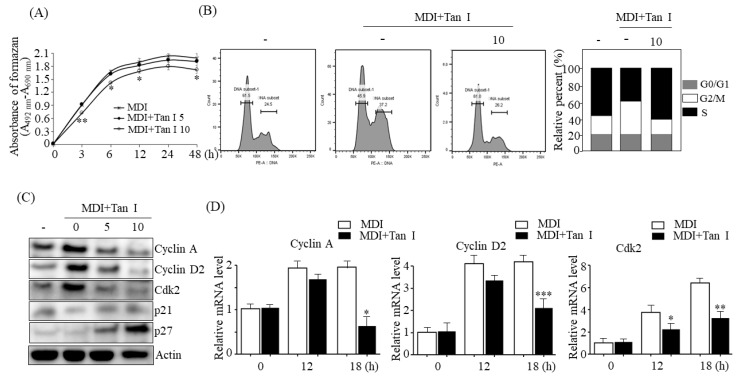
Tan I suppressed mitotic clonal expansion (MCE) during early adipogenesis. The 3T3-L1 preadipocytes were differentiated in the MDI with 10 μM Tan I for indicated times. (**A**) Cell numbers were measured at indicated times. (**B**) The cells were harvested with PI. The stained cells were analyzed by flow cytometer, and the cell population at each stage of the cell cycle was determined by BD Pro software. (**C**) The protein levels of cyclin A, cyclin D2, Cdk2, p21, and p27 were assessed by western blotting. The full length western blots are provided in Appendix A. (**D**) The mRNA levels of *cyclin A*, *cyclin D2*, and *Cdk2* were measured by qPCR. The data are presented as the mean ± SEM of three replicate experiments. * *p* < 0.05, ** *p* < 0.01, *** *p* < 0.001 vs. differentiated 3T3-L1 adipocytes without Tan I treatment. MDI means differentiation medium.

**Figure 5 nutrients-12-01242-f005:**
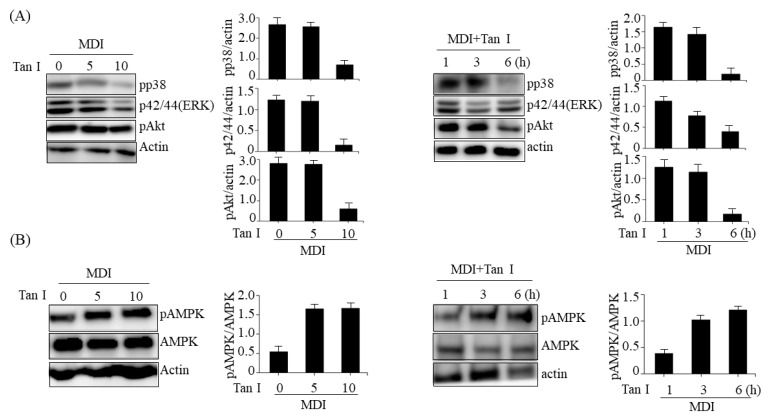
Tan I inhibited p38, extracellular signal-regulated kinase (ERK), PI3/Akt, and activated AMP-activated protein kinase (AMPK) during early adipogenesis. The 3T3-L1 preadipocytes were differentiated in the MDI with 5 or 10 μM Tan I for 6 h (left) or differentiated in the MDI with 10 μM Tan I for indicated times (right). The levels of pp38, p42/44 (Erk), pAkt (**A**), and pAMPK (**B**) were analyzed by western blotting. The full length western blots are provided in Appendix A. Bar graph represents densitometric analysis of band intensity ratio for pp38/Actin, p42/44 (Erk)/Actin, pAkt/ Actin, and pAMPK/AMPK. MDI means differentiation medium.

**Figure 6 nutrients-12-01242-f006:**
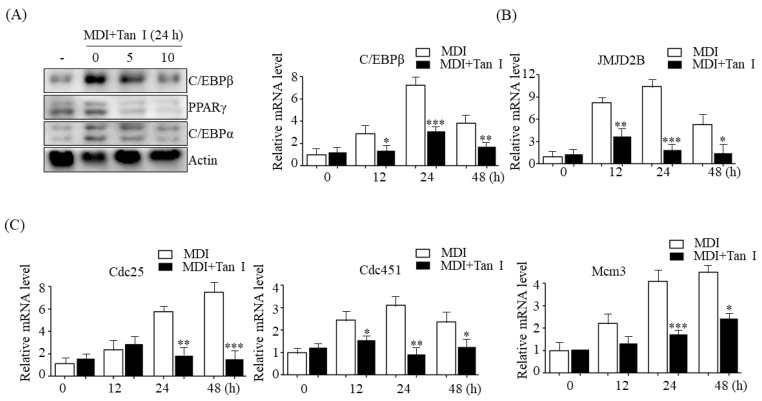
Tan I inhibited expression of C/EBPβ, JMJD2B, and cell cycle genes during early adipogenesis. The 3T3-L1 preadipocytes were differentiated in the MDI with 5 or 10 μM Tan I for 24 h or differentiated in the MDI with 10 μM Tan I for indicated times. (**A**) The protein levels of C/EBPβ, PPARγ, and C/EBPα were determined by western blotting. The full length western blots are provided in Appendix A. The mRNA levels of (**B**) *C/EBPβ*, *JMJD2B* and (**C**) *Cdc25*, *Cdc45l*, and *Mcm3* were measured by qPCR. The data are presented as the mean ± SE of three replicate experiments. * *p* < 0.05, ** *p* < 0.01, *** *p* < 0.001 vs. 3T3-L1 adipocytes without Tan I treatment at indicated time. MDI means differentiation medium.

**Figure 7 nutrients-12-01242-f007:**
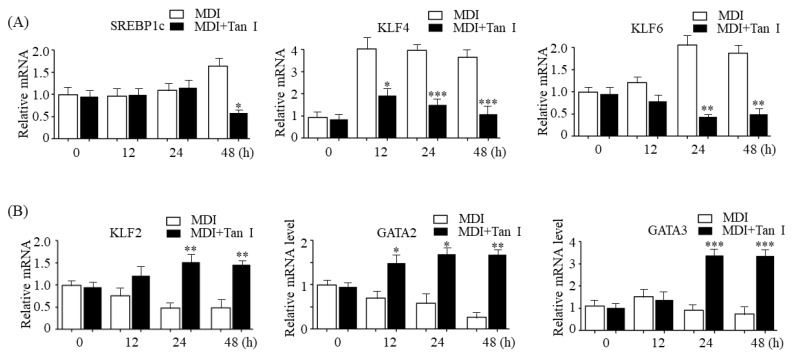
Tan I regulated expression of early adipogenic transcription factors during early adipogenesis. The 3T3-L1 preadipocytes were differentiated in the MDI with 10 μM Tan I for indicated times. (**A**) mRNA levels of early adipogenic activators including *SREBP1c*, *KLF4,* and *KLF6* were determined by qPCR. (**B**) mRNA levels of early adipogenic repressors including *KLF2*, *GATA2,* and *GATA3* were determined by qPCR. The data are presented as the mean ± SEM of three replicate experiments. * *p* < 0.05, ** *p* < 0.01, *** *p* < 0.001 vs. 3T3-L1 adipocytes without Tan I treatment at indicated time.

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
