# Peer review of "Anti-Obesity Effects of Tanshinone I from Salvia miltiorrhiza Bunge in Mice Fed a High-Fat Diet through Inhibition of Early Adipogenesis"

_nutrients, 2020, doi:10.3390/nu12051242_

Round 1

Reviewer 1 Report

The research is well organized and the manuscript writing is good. The plagiarism detection has shown that a relatively high similarity was found with previous publication (Scientific Reports | 7:40345). Subsequently, the manuscript can be accepted after a slight modification.

Author Response

Response to reviewer' comments

- Thank you very much for your careful checking. The plagiarism has been modified slightly according to your suggestion. The modification is shown in red fonts in revised manuscript.

Reviewer 2 Report

Jung et al. reported the anti-obesity effects of Tanshinone I on mice fed with a high fat diet, followed by investigating the related molecular mechanism in 3T3-L1 cells. The results showed that Tan I-mediate anti-obesity was resulted from the inhibition of early adipogenesis. This manuscript was written well, easy to read and presented clearly, and quite a lot of data were presented to support the conclusion obtained. Also, this study is interesting, useful and imply that Tan I as a potential therapy for the treatment of obesity-associated diseases.

There are several major concerns and several minor comments must to taken into consideration.

Major concerns:

  1. Tan I is a lipid-soluble compound, but in line 95, the authors mentioned that “Tan I was dissolved in distilled water”, this can be a big issue for all the following experiments performed in this study. Tan I either can not dissolve in water or has very low solubility in water, then the concentrations of Tan I used in all the experiments in this study can be in-correct. Tan I from Sigma suggest that Tan I is soluble in DMSO at 1 mg/ml, then the control should be used by applying the same amount of DMSO.

Response:
We appreciate your critical pointing. As your pointing, we made a mistake of the statement in line 95 that Tan I was dissolved in distilled water. Correctly, Tan I was dissolved in DMSO at 1 mg/ml and diluted in distilled water. Furthermore, the mixtures of DMSO and water were used for control. Thus, we have corrected the statement in line 95 to the statement that “Tan I was dissolved in DMSO at 1 mg/ml, diluted in distilled water.”(line 95)

  1. By searching the literature, it was noted that Tan IIA has been reported to show anti-adipogenic effects on 3T3-L1 (Park et al., 2019), but this highly-related study was ignored in this manuscript. All the related studies including Park et al., 2019 must be mentioned, discussed and compared with the current results obtained in this work.

Response:
Thank you for your valuable comments. Besides Tan I, two compounds belonging to the
group of tanshinone including tanshinone IIA and cryptotanshinone have been reported to have anti-adipogenic effects. They exerted an inhibitive effect on adipogenesis of 3T3-L1 through downregulation of positive adipogenesis factors including C/EBPα, C/EBPβ and PPARγ, upregulation of negative adipogenesis factors GATA2, CHOP and TNF-α, and inhibition of STAT3/ 5 signaling.
Our current study also revealed that Tan I repressed the expression of early adipogenic activators, C/EBPβ, SREBP1c, KLF4, KLF6, whereas increased the expression of early adipogenic repressors, KLF2, GATA2 and GATA3 shown in Fig 7, whose results were similar to those of two compounds.
However, we observed additionally that Tan I inhibited the adipogenesis of 3T3-L1 through the attenuation of mitotic clonal expansion via cell cycle arrest at the G1/S phase transition, and repression of histone demethylase JMJD2B associated with upregulation of cell cycle genes. Furthermore, our current study demonstrated that Tan I inhibited the phosphorylation of p38, ERK, and Akt during the process of MCE, while stimulated the phosphorylation of AMP-activated protein kinase.
Thus, the important difference in molecular mechanisms between Tan I and other two compounds is that Tan 1 attenuates mitotic clonal expansion during early adipogenesis, and represses the expression of epigenetic regulator, JMJD2B associated with activation of adipogenesis, which may also contribute to the inhibition of adipogenesis of 3T3-L1.
Furthermore, we evaluated anti-obese effects in high fat diet (HFD)-induced obese mice in vivo. Tan I significantly prevented HFD-induced obesity, and improved glucose utilization and insulin sensitivity. However, the anti-obese effects of two compounds including tanshinone IIA and cryptotanshinone have not been verified in obese animal model.
We have mentioned the studies of two compounds in Discussion (lane 429-436) and added them in Reference (reference 16 & 17).

References:
(1) Park, Y.K.; Obiang-Obounou, B.W.; Lee, J.; Lee, T.Y.; Bae, M.A.; Hwang, K.S.; Lee, K.B.; Choi, J.S.; Jang, B.C. Anti-Adipogenic Effects on 3T3-L1 Cells and Zebrafish by Tanshinone IIA. Int J Mol Sci. 2017, 18, 2065
(2) Rahman, N.; Jeon, M.; Song, H.Y.; Kim, Y.S. Cryptotanshinone, a compound of Salvia miltiorrhiza inhibits pre-adipocytes differentiation by regulation of adipogenesis-related genes expression via STAT3 signaling. Phytomedicine. 2016, 23, 58-67

  1. Given the similar and related studies mentioned above, it is important to clarify what is the novelty of this work, and what are the similarities and differences. Which compound is better

Response:
As mentioned in the response to the Comment 2, the important difference in molecular mechanisms between Tan I and other two compounds including tanshinone IIA and cryptotanshinone is that Tan 1 attenuates mitotic clonal expansion during early adipogenesis, and represses the expression of epigenetic regulator, JMJD2B associated with activation of adipogenesis, which may contribute to the inhibition of adipogenesis of 3T3-L1.
Furthermore, we evaluated anti-obese effects in high fat diet (HFD)-induced obese mice in vivo. Tan I significantly prevented HFD-induced obesity, and improved glucose utilization and insulin sensitivity. However, the anti-obese effects of two compounds have not been verified in obese animal model.

  1. Are there other compounds belong to the group of tanshinone show anti-adipogenic effects?

Response:
As mentioned in the response to the Comment 2, tanshinone IIA and cryptotanshinone have been reported to exert anti-adipogenic effects.

  1. For other compounds reported to show anti-adipogenic effects, how they are working and are they better than Tan I or not. Advantages and disadvantages?This should be discussed more.       

Response:
As mentioned in the response to the Comment 2, tanshinone IIA and cryptotanshinone exerted an inhibitive effect on adipogenesis of 3T3-L1 through downregulation of positive adipogenesis factors including C/EBPα, C/EBPβ and PPARγ, upregulation of negative adipogenesis factors GATA2, CHOP and TNF-α, and inhibition of STAT3/ 5 signaling.
Compared with tanshinone IIA and cryptotanshinone, Tan I have a diverse anti-adipogenic effect as mentioned in the response to Comment 2.  

Minor comments:

Lines 144-146, whether all the details including information for all antibodies of different proteins for western blot are all included in [7]. If not, more information should be provided here.                                                                                                                                                                                                 

The information for all antibodies of different proteins for western blot was described in Materials and methods (line 83-85).                           

  1. How the concentrations of Tan I used in mice and 3T3-L1 cells were selected? Based on published papers or screened first? If they are chosen based on published papers, related papers should be cited and described. If they are selected by pre-screening, the related results should be described.                                                                                                                                                                                                                     For animal study using C57BL/6 mice, we determined the concentration of Tan I based on the previous study to see the effects of Tan I in a mouse mode of Parkinson’s disease using C57BL/6 mice. 5 mg/kg of body weight and 10 mg/kg of body weight were used in the previous study, whereas 2 mg/kg and 5 mg/kg were used in our current study.
    For in vitro study using 3T3-L1 cells, we determined the concentration of Tan I by XTT assay. The concentration up to 10 μM Tan I produced no inhibitory effects on the viability of 3T3-L1 cells. Thus, 5 μM and 10 μM Tan I were used for in vitro study.
    Reference:
    Wang S, Jing H, Yang H, Liu Z, Guo H, Chai L, Hu L. Tanshinone I selectively suppresses pro-inflammatory genes expression in activated microglia and prevents nigrostriatal dopaminergic neurodegeneration in a mouse model of Parkinson's disease. J Ethnopharmacol. 2015, 164:247-55.                                                                                                 
  2. Line 272, revise “Tan1” to “Tan I”.                                                                                                                                                                                                                                                                                 We corrected “Tan1” to “Tan I”.

Explain why the expression of Cycin A and D2 only reduced at 18 h but not 12 h after MDI and Tan I treatment.                                                                                                                                                                                 The expression of cyclin A and D2 began to reduce slightly from 12 h.

In summary, I think this manuscript and only be accepted after all the above comments are addressed.  

Round 2

Reviewer 2 Report

The authors have responded to all of my comments and revisions have been properly made. I have no further comments.